# GraphCGAN: Convolutional Graph Neural Network with Generative Adversarial Networks

## Abstract

Graph convolutional networks (GCN) achieved superior performances in graph-based semi-supervised learning (SSL) tasks. Generative adversarial networks (GAN) also show the ability to increase the performance in SSL. However, there is still no good way to combine the GAN and GCN in graph-based SSL tasks. In this work, we present GraphCGAN, a novel framework to incorporate adversarial learning with convolution-based graph neural network, to operate on graph-structured data. In GraphCGAN, we show that generator can generate topology structure and attributes/features of fake nodes jointly and boost the performance of convolution-based graph neural network classifier. In a number of experiments on benchmark datasets, we show that the proposed GraphCGAN outperforms the reference methods by a significant margin.

## 1 Introduction

Graph-based semi-supervised learning (SSL) aims to classify nodes in graph, where only small amounts of nodes are labeled due to the expensive and time-consuming label collection process. To solve such task, various graph neural networks (GNNs) have been proposed using the idea of convolutional neural networks (CNN) to implicitly propagate the information of labeled nodes to unlabeled nodes through the linkage between nodes (Kipf & Welling, 2016; Veličković et al., 2017; Hamilton et al., 2017). These convolution-based graph neural networks have achieved superior performance on multiple benchmark datasets in graph-based SSL tasks (Wu et al., 2019).

Recently, generative adversarial networks (GANs) (Goodfellow et al., 2014) have been shown a power in improving the performance of image-based SSL problems (Odena, 2016; Salimans et al., 2016; Li et al., 2019b). In semi-GAN (Salimans et al., 2016), authors converted the $M$-class classification task into solving $(M + 1)$-class problem where the synthetic $(M + 1)$th class is generated by the GAN's generator. Later on, Dai et al. provided a theoretical insight that the generated data are able to boost the performance of classifier under certain assumptions. Our work is motivated by the the semi-GAN.

GraphSGAN (Ding et al., 2018) first investigated the adversarial learning over graph, where the graph is embedding into an embedding space and synthetic data are generated in the corresponding space. The multi-layer perceptron (MLP) is trained as the classifier on the embedding vectors. However, to our knowledge, there is still no existed method to combine the adversarial learning to convolution-based GNNs on graph-based SSL task. In this work, we explore the potential of incorporating the convolution-based GNN and GAN. The challenges of constructing a general framework have three folds: first, the attributed graph data are non-Euclidean whose distribution contains information of graph topology structure as well as the attributes of nodes. Hence, it is not trivial to construct generator to model the distribution. Second, even the generator can model the graph's distribution, the generator should be trained properly to boost the performance of the classifier. A poor-quality generator would introduce noise to the existed graph and affect the classifier. Third, many variants of GCN have been proposed continuously. The framework should be built with flexibility to adapt to different convolution-based GNNs.

We construct a novel approach called GraphCGAN to deal with above challenges. First, to model the distribution of graph, the generator is built sequentially from two sub-generators: one models

the attribute information (node's attribute) and another one models the graph topology structure (adjacency relation of node). Details can be found in Section 3.1. Second, in GraphCGAN, the generator is trained based on the feature matching technique (Salimans et al., 2016) which minimizes the distance between generated nodes and real nodes in the constructed feature space. This technique showed a good performance in SSL tasks in practice. The details for construction of loss functions can be found in Section 3.3. For GCN, the attributes of nodes are aggregated convolutionally by multiple layers. The representation of the last layer is usually considered as the prediction for the labels. For variants of GCN, the main differences exist in the strategy of layer aggregation (Hamilton et al., 2017). In our framework, we choose the second to the last layer of convolution-based GNN as the feature matching functions. Therefore, our framework is easily extended to variants of GCN. More discussions can be found in Section 3.2.

## 2 PRELIMINARY

We first introduce the notation about graph. Let $\mathcal{G} = (V, E)$ denote a graph, where $V$ is the set of nodes with $|V| = n$ and $E \subset V \times V$ is a set of edges with $|E| = m$. The adjacency matrix $\mathbf{A} \in \mathbb{R}^{|V| \times |V|}$ is defined as $A_{ij} = 1$ if node $v_i$ and $v_j$ has edge, otherwise $A_{ij} = 0$. Suppose each node $v_i$ has a d-dimensional feature $\mathbf{x}_i \in \mathbb{R}^d$ and a single value label $y_i \in \{1, 2, .., M\}$. In the semi-supervised learning setting, there is a disjoint partition for the nodes, $V = V^{\mathcal{L}} \cup V^{\mathcal{U}}$, such that, for $v_i \in V^{\mathcal{L}}$, the corresponding label is known and for $v_j \in V^{\mathcal{U}}$ the corresponding label is unknown. The distributions of node in labeled set $V^{\mathcal{L}}$ and unlabeled set $V^{\mathcal{U}}$ are denoted as $p_{V^{\mathcal{L}}}$ and $p_{V^u}$, respectively. The semi-supervised learning is to learn the label for unlabeled set $\{y_j | v_j \in V^{\mathcal{U}}\}$ given adjacency matrix $\mathbf{A}$, feature matrix $\mathbf{X} = [\mathbf{x}_i]_{v_i \in V}$ and labels for labeled sets $\{y_i | v_i \in V^{\mathcal{L}}\}$.

### 2.1 CONVOLUTION BASED GRAPH NEURAL NETWORK CLASSIFIER

Based on the Laplacian smoothing, the convolution-based GNN models propagate the information of nodes features across the nodes' neighbors in each layer. Specifically, in GCN, the layer-wise propagation rule can be defined as follows:

$$\mathbf{H}^{(l+1)} = \sigma(\mathbf{D}^{-1}\mathbf{A}\mathbf{H}^{(l)}\mathbf{W}^{(l)} + \mathbf{b}^{(l)}), \ l = 0, 1, 2.., L-1 \tag{1}$$

where $\mathbf{W}^{(l)}$ and $\mathbf{b}^{(l)}$ are layer-specific trainable weight matrix and bias, respectively. $\sigma(\cdot)$ is an activation function. $\mathbf{D}$ is the diagonal degree matrix with $D_{ii} = \sum_j A_{ij}$. Hence, $\mathbf{D}^{-1}\mathbf{A}$ represents normalization of adjacency matrix $\mathbf{A}$. The initial layer $\mathbf{H}^{(0)}$ is the feature matrix $\mathbf{X}$. The final layer $\mathbf{H}^{(L)}$ followed by a $soft \max$ layer can be viewed as the prediction of one-hot representation for the true label $\mathbf{y}$.

Recently, many variants of the GCN layer-wise propagation rule had been proposed, including graph attention network, cluster GCN (Veličković et al., 2017; Chiang et al., 2019), which achieved state-of-the-art performances in many benchmark datasets.

### 2.2 GENERATIVE ADVERSARIAL NETWORK BASED SEMI-SUPERVISED LEARNING

In semi-GAN, the classifier $C$ and generator $G$ play a non-cooperative game, where classifier aims to classify the unlabeled data as well as distinguish the generated data from real data; generator attempts to match feature of real data and that of generated data. Therefore, the objective function for classifier can be divided into two parts (Salimans et al., 2016). The first part is the supervised loss function

$$\mathcal{L}_{sup} = \mathbb{E}_{v,y \sim p_{V\mathcal{L}}} \log P_C(y|v, y \leq M)$$

which is the log probability of the node label given the real nodes. The second part is the unsupervised loss function

$$\mathcal{L}_{un-sup} = \mathbb{E}_{v \sim p_{V^u}} \log P_C(y \leq M|v) + \mathbb{E}_{v \sim p_{V^G}} \log P_C(y = M + 1|v)$$

which is the sum of log probability of the first M classes for real nodes and the log probability of the (M + 1)th class for generated nodes $V^G$. The classifier C can be trained by maximize the objective function

$$\mathcal{L}_C = \mathcal{L}_{sup} + \mathcal{L}_{un-sup}. \tag{2}$$

For objective function of generator, Salimans et al. (2016) found minimizing feature matching loss in Equation 3 achieved superior performance in practice

$$\mathcal{L}_G = ||\mathbb{E}_{v \sim p_{V \cup U}}(\mathbf{f}(v)) - \mathbb{E}_{\mathbf{z} \sim p_{\mathbf{z}}(\mathbf{z})}(\mathbf{f}(G(\mathbf{z})))||_2^2, \tag{3}$$

where the feature matching function $\mathbf{f}(\cdot)$ maps the input into a feature space and $\mathbf{z} \sim p_{\mathbf{z}}(\mathbf{z})$ is drawn from a given distribution like uniform distribution. Furthermore, Dai et al. (2017) provided a theoretical justification that complementary generator G was able to boost the performance of classifier C in SSL task.

## 3 FRAMEWORK OF GRAPHCGAN

To combine the aforementioned Laplacian smoothing on graph and semi-GAN on SSL together, we develop GraphCGAN model, using generated nodes to boost the performance of convolution-based GNN models.

### 3.1 CONSTRUCTION OF GENERATOR FOR GRAPHCGAN

The generator G generates fake node $v_0$ by generating feature vector $\mathbf{x}_0 \in \mathbb{R}^d$ and adjacency relation $\mathbf{a}_0 \in \mathbb{R}^n$ jointly, where $a_{0,i} = 1$ if the fake node is connected to real node $v_i$, otherwise $a_{0,i} = 0$. Therefore, the distribution for generated node $p_G(v_0)$ can be expressed by the joint distribution of the corresponding feature and adjacency relation $p_G(\mathbf{x}_0, \mathbf{a}_0)$. From the conditional distribution formula, the joint distribution can be written as $p_G(\mathbf{x}_0, \mathbf{a}_0) = p_{G_1}(\mathbf{x}_0)p_{G_2}(\mathbf{a}_0|\mathbf{x}_0)$. We use sub-generators $G_1$ and $G_2$ to generate fake feature $\mathbf{x}_0$ and $\mathbf{a}_0|\mathbf{x}_0$, respectively. In practice, $\mathbf{a}_0|\mathbf{x}_0$ can be modeled by $G_2(\mathbf{z}; \mathbf{x}_0) = G_2(\mathbf{z}; G_1(\mathbf{z}))$ where the adjacency relation $\mathbf{a}_0$ is constructed by sub-generator $G_2$ given the input of $\mathbf{x}_0$. The distribution of generated node can be denoted by

$$p_G(v_0) = p_G(\mathbf{x}_0, \mathbf{a}_0) = p_G(\mathbf{x}_0)p(\mathbf{a}_0|\mathbf{x}_0) = p(G_1(\mathbf{z}))p(G_2(\mathbf{z}; G_1(\mathbf{z}))) =: p(G(\mathbf{z})). \tag{4}$$

If $B$ nodes $(v_{0,1}, v_{0,2}, .., v_{0,B})$ are generated, the generated feature matrix is denoted as $\mathbf{X}_0 = (\mathbf{x}_{0,1}^T, \mathbf{x}_{0,2}^T, .., \mathbf{x}_{0,B}^T)^T$ and generated adjacency matrix has form $\mathbf{A}_0 = (\mathbf{a}_{0,1}^T, \mathbf{a}_{0,2}^T, .., \mathbf{a}_{0,B}^T)^T$. Hence, the combined adjacency matrix can be denoted as

$$\tilde{\mathbf{A}} = \begin{bmatrix} \mathbf{A} & \mathbf{A}_0^T \\ \mathbf{A}_0 & \mathbf{I}_B \end{bmatrix} \in \mathbb{R}^{(n+B) \times (n+B)}, \tag{5}$$

The combined feature vector is

$$\tilde{\mathbf{X}} = \begin{bmatrix} \mathbf{X} \\ \mathbf{X}_0 \end{bmatrix} \in \mathbb{R}^{(n+B) \times d}. \tag{6}$$

The diagonal degree matrix $\tilde{\mathbf{D}} \in \mathbb{R}^{(n+B) \times (n+B)}$ can be denoted as $\begin{bmatrix} \mathbf{D}_* & \mathbf{0} \\ \mathbf{0} & \mathbf{D}_B \end{bmatrix}$ where $\mathbf{D}_* \in \mathbb{R}^{n \times n}$ with $D_{*,ii} = \sum_j A_{ij} + \sum_b A_{0,bi}$ and $\mathbf{D}_B \in \mathbb{R}^{B \times B}$ with $D_{B,bb} = \sum_j A_{0,bj} + 1$.

### 3.2 ANALYSIS OF CLASSIFIER FOR GRAPHCGAN

In GraphCGAN, we adopt the convolution-based GNN, such as GCN, GraphSage (Hamilton et al., 2017) or GAT (Veličković et al., 2017), as the the classifier. The classifier is applied to the enlarged graph $\tilde{\mathcal{G}} = [\tilde{\mathbf{X}}, \tilde{\mathbf{A}}]$ to obtain the prediction $\tilde{\mathbf{y}}$ of nodes $V \cup V^G$.

Specially, considering the layer-wise propagation of GCN (Equation 1) as the classifier in GraphC-GAN, the propogation rule can be denoted as

$$
\begin{aligned}
\tilde{\mathbf{H}}^{(l+1)} &= \sigma(\tilde{\mathbf{D}}^{-1}\tilde{\mathbf{A}}\tilde{\mathbf{H}}^{(l)}\mathbf{W}^{(l)} + \tilde{\mathbf{b}}^{(l)}) \\
&= \sigma(\begin{bmatrix} \mathbf{D}_*^{-1} & \mathbf{0} \\ \mathbf{0} & \mathbf{D}_B^{-1} \end{bmatrix}\begin{bmatrix} \mathbf{A} & \mathbf{A}_0^T \\ \mathbf{A}_0 & \mathbf{I}_B \end{bmatrix}\begin{bmatrix} \mathbf{H}_*^{(l)} \\ \mathbf{H}_0^{(l)} \end{bmatrix}\mathbf{W}^{(l)} + \begin{bmatrix} \mathbf{b}^{(l)} \\ \mathbf{b}_B^{(l)} \end{bmatrix}) \\
&= \sigma(\begin{bmatrix} \mathbf{D}_*^{-1}\mathbf{A}\mathbf{H}_*^{(l)} + \mathbf{D}_*^{-1}\mathbf{A}_0^T\mathbf{H}_0^{(l)} \\ \mathbf{D}_B^{-1}\mathbf{A}_0\mathbf{H}_*^{(l)} + \mathbf{D}_B^{-1}\mathbf{H}_0^{(l)} \end{bmatrix}\mathbf{W}^{(l)} + \begin{bmatrix} \mathbf{b}^{(l)} \\ \mathbf{b}_B^{(l)} \end{bmatrix}) \\
&= \sigma(\begin{bmatrix} \mathbf{D}_*^{-1}\mathbf{A}\mathbf{H}_*^{(l)}\mathbf{W}^{(l)} + \mathbf{b}_*^{(l)} \\ (\mathbf{D}_B^{-1}\mathbf{A}_0\mathbf{H}_*^{(l)} + \mathbf{D}_B^{-1}\mathbf{W}^{(l)})\mathbf{W}^{(l)} + \mathbf{b}_B^{(l)} \end{bmatrix}) \\
&=: \begin{bmatrix} \mathbf{H}_*^{(l+1)} \\ \mathbf{H}_0^{(l+1)} \end{bmatrix}.
\end{aligned}
\tag{7}
$$

where the first layer is chosen as the enlarged feature matrix $\tilde{\mathbf{H}}^{(0)} = \tilde{\mathbf{X}}$. Weight matrix $\mathbf{W}^{(l)}$ has the same in Equation 1. Bias vector $\tilde{\mathbf{b}}^{(l)}$ has dimension $(n + B)$ which is denoted as $[\mathbf{b}^{(l)T}, \mathbf{b}_B^{(l)T}]^T$. We denote $\mathbf{b}_*^{(l)} = \mathbf{D}_*^{-1}\mathbf{A}_0^T\mathbf{H}_*^{(l)}\mathbf{W}^{(l)} + \mathbf{b}^{(l)}$ to make the format clear. From Equation 7, the layer propagation of real nodes (first n rows) follows the same format as the GCN layer propagation in Equation 1. As a special case, for the zero generator $\mathbf{A}_0 = \mathbf{0}$ or $\mathbf{X}_0 = \mathbf{0}$, the performance of classifier on $V \cup V^G$ would be the same as that of original classifier on $V$.

For the last layer $\tilde{\mathbf{H}}^{(L)} \in \mathbb{R}^{(n+B)\times M}$, we adopt the strategy in Salimans et al. (2016) to obtain the $(M + 1)$ class label $\tilde{\mathbf{y}}$ by

$$
\tilde{\mathbf{y}} = soft\max(\tilde{\mathbf{H}}^{(L)}||\mathbf{0}_{(n+B)\times 1}),
\tag{8}
$$

where $||$ denotes concatenation and $\mathbf{0}_{(n+B)\times 1} \in \mathbb{R}^{(n+B)\times 1}$ is a zero matrix. The loss function for classifier in GraphCGAN follows the same format in Equation 2.

### 3.3 LOSS FUNCTIONS

Let us denote $\mathbf{g}(\cdot, \cdot; \theta_C)$ as the map from feature vector and adjacency vector to the space of second to the last layer in convolution-based GNN with trainable parameter $\theta_C$. Specially, in the case of GCN, for node $v_i$ with feature vector $\mathbf{x}_i$ and adjacency vector $\mathbf{a}_i$,

$$
\mathbf{g}(\mathbf{x}_i, \mathbf{a}_i; \theta_C) = \tilde{\mathbf{H}}_i^{(L-1)},
\tag{9}
$$

where $\tilde{\mathbf{H}}_i^{(L-1)}$ denotes the i-th row of $\tilde{\mathbf{H}}^{(L-1)}$ and $\theta_C = [\mathbf{W}^{(l)}; \tilde{\mathbf{b}}^{(l)}]_{l=0,1,..,L-2}$.

According to Equation 4, the loss function of generator $G$ can be decomposed into two parts: the loss functions of sub-generators $G_1$ and $G_2$ separately. To construct $G_1$, the feature matching function $\mathbf{f}$ in Equation 3 should solely depend on feature vector. Therefore, we mask the adjacency matrix $\tilde{\mathbf{A}}$ as identity matrix $\tilde{\mathbf{I}} \in \mathbb{R}^{(n+B)\times(n+B)}$ in layer propagation. Formally, the feature matching loss function of $G_1$ is constructed as

$$
\mathcal{L}_{G_1} = ||\mathbb{E}_{\mathbf{x}_i}(\mathbf{g}(\mathbf{x}_i, \mathbf{I}_i; \theta_C)) - \mathbb{E}_{\mathbf{z}\sim p_\mathbf{z}(\mathbf{z})}(\mathbf{g}(G_1(\mathbf{z}), \mathbf{0}; \theta_C))||_2^2,
$$

where $\mathbf{I}_i$ denote the i-th row of identity matrix $\mathbf{I} \in \mathbb{R}^{n\times n}$ and $\mathbf{0}$ is the zero vector.

After $\mathbf{x}_0 = G_1(\mathbf{z})$ is built, the feature matching loss function of $G_2$ can be constructed similarly from

$$
\mathcal{L}_{G_2} = ||\mathbb{E}_{\mathbf{a}_i}(\mathbf{g}(\mathbf{x}_i, \mathbf{a}_i; \theta_C)) - \mathbb{E}_{\mathbf{z}\sim p_\mathbf{z}(\mathbf{z})}(\mathbf{g}(\mathbf{x}_0, G_2(\mathbf{z}); \theta_C))||_2^2.
$$

Therefore, loss function for $G$ can be written as

$$
\mathcal{L}_G = \mathcal{L}_{G_1} + \mathcal{L}_{G_2}.
\tag{10}
$$

Furthermore, when multiple fake nodes are generated, Salimans et al. (2016) showed that adding pull-away item to loss function can increase the entropy of generator which led to better performance

in practice. The pull-away loss for sub-generators $G_1$, $G_2$ can be denoted as

$$\mathcal{L}_{G_1}^{pt} = \frac{1}{B(B-1)} \sum_i^B \sum_{j \neq i} \left( \frac{\mathbf{g}(G_1(\mathbf{z}_i), \mathbf{0}; \theta_C)^T \mathbf{g}(G_1(\mathbf{z}_j)), \mathbf{0}; \theta_C)}{||\mathbf{g}(G_1(\mathbf{z}_i)), \mathbf{0}; \theta_C)|| ||\mathbf{g}(G_1(\mathbf{z}_j)), \mathbf{0}; \theta_C)||} \right)$$

and

$$\mathcal{L}_{G_2}^{pt} = \frac{1}{B(B-1)} \sum_i^B \sum_{j \neq i} \left( \frac{\mathbf{g}(\mathbf{x}_{0,i}, G_2(\mathbf{z}_i); \theta_C)^T \mathbf{g}(\mathbf{x}_{0,j}, G_2(\mathbf{z}_j); \theta_C)}{||\mathbf{g}(\mathbf{x}_{0,i}, G_2(\mathbf{z}_i); \theta_C)|| ||\mathbf{g}(\mathbf{x}_{0,j}, G_2(\mathbf{z}_j); \theta_C)||} \right).$$

The loss function for G with pull-away item can be written as

$$\mathcal{L}_G^* = \mathcal{L}_G + \mathcal{L}_{G_1}^{pt} + \mathcal{L}_{G_2}^{pt}. \tag{11}$$

Besides, Dai et al. (2017) constructed the complementary loss by

$$\mathcal{L}_{G_1}^c = \mathbb{E}_{\mathbf{x} \sim p_{G_1}} \log(p(\mathbf{x})) \mathbb{I}(p(\mathbf{x}) > \varepsilon), \quad \mathcal{L}_{G_2}^c = \mathbb{E}_{\mathbf{a} \sim p_{G_2}} \log(p(\mathbf{a})) \mathbb{I}(p(\mathbf{a}) > \varepsilon),$$

which could also increase performance. Therefore, the loss function for G with complementary loss can be written as

$$\mathcal{L}_G^{**} = \mathcal{L}_G^* + \mathcal{L}_{G_1}^c + \mathcal{L}_{G_2}^c. \tag{12}$$

The procedure is formally presented in Algorithm 1.

---

**Algorithm 1:** GraphCGAN Algorithm

---

**Input:** Adjacency matrix $\mathbf{A}$, Node feature $\mathbf{X}$, initialized fake nodes $V^G = [\mathbf{A}_0, \mathbf{X}_0]$.
   hyper-parameters including dimension of the noise vector $d_{noise}$, the number of steps
   $K_D$, and the size of fake nodes $B$ and early stop error.
**Output:** Prediction $\tilde{\mathbf{Y}}$
1 **while** *not early stop* **do**
2   Combine the fake nodes $V^G$ to the graph and obtain $\tilde{\mathbf{A}}$ and $\tilde{\mathbf{X}}$ from Equation 5 and
   Equation 6;
3   **Classifier:**
4   $iter_D = 0$
5   **while** $iter_D < K_D$ **do**
6    Use convolution-based GNN as the classifier C, and extract the map to the intermediate
    layer $\mathbf{g}(.,.)$ as Equation 9;
7    Train C by minimizing $\mathcal{L}_C$ (Equation 2) on combined graph, obtain predicted result $\tilde{\mathbf{Y}}$;
    $iter_D = iter_D + 1$;
8   **Generator:**
9   Generate a random noise vector $\mathbf{Z} \sim U(\mathbf{0}, \mathbf{I}) \in \mathbb{R}^{B \times d_{noise}}$;
10   Train generator $G = [G_1; G_2]$ by minimizing Equation 10 or Equation 11 or Equation 12;
11   Obtain $\mathbf{X}_0 = G_1(\mathbf{Z})$ and $\mathbf{A}_0 = G_2(\mathbf{Z}; G_1(\mathbf{Z}))$.

---

## 4 RELATED WORK

### 4.1 GRAPH-BASED SEMI-SUPERVISED LEARNING

The challenge for graph-based SSL is to leverage unlabeled data to improve performance in classification. There are three categories of the Graph-based semi-supervised learning. The first one is the Laplacian regularization-based methods (Xiaojin & Zoubin, 2002; Lu & Getoor, 2003; Belkin et al., 2006). The second type is the embedding-based methods, including DeepWalk (Perozzi et al., 2014), SemiEmb (Weston et al., 2012), and Planetoid (Yang et al., 2016). The third type is convolutional based graph neural networks such as GCN (Kipf & Welling, 2016), GAT (Veličković et al., 2017), ClusterGCN (Chiang et al., 2019) and DeepGCN (Li et al., 2019a). Such methods address the semi-supervised learning in an end-to-end manner. Convolution-based methods perform the graph convolution by taking the weighted average of a node's neighborhood information. In many graph semi-supervised learning tasks, the convolution-based methods achieved the state-of-the-art performance (Wu et al., 2019).

## 4.2 GNN LEARNING WITH GAN

GAN is wildly used in obtaining generative graph models. GraphGAN (Wang et al., 2018) proposed a framework for graph embedding task. Specifically, GraphGAN can generate the link relation for a center node. However, GraphGAN cannot be applied to attributed graph.

MolGAN (De Cao & Kipf, 2018) proposed a framework for generating the attributed graph of molecule by generating the adjacency matrix and feature matrix independently. After that, MolGAN used an the score for the generated molecule as reward function to choose the reasonable combination of attributes and topology structure by an auxiliary reinforcement learning model. In comparison, GraphCGAN can generate attributes and adjacency matrix of the attributed graph jointly, which can capture the correlation between the attributes and topology relation.

DGI (Veličković et al., 2018) proposed a general approach for learning node representations within graph-structured data in an unsupervised manner. For the generator, In DGI, the fake nodes are created from a pre-specified corruption function applied on the original nodes. In contrast, our GraphCGAN can generate the fake nodes from a dynamic generator during the training GAN process. For the classifier, the DGI uses GCN only, however, our GraphCGAN is flexible and adaptive to other convolution-based GNN models.

## 4.3 GAN WITH SEMI-SUPERVISED LEARNING

SGAN (Odena, 2016) first introduced the adversarial learning to the semi-supervised learning on image classification task. GAN-FM (Salimans et al., 2016) stabilized training process in SGAN by introducing feature-matching and minibatch techniques. In Kumar et al. (2017), authors discuss about the effects of adding fake samples and claimed that moderate fake samples could improve the performance in image classification task.

GraphSGAN Ding et al. (2018) proposed a framework for graph Laplacian regularization based classifier with GAN to solve graph-based semi-supervised learning tasks. In GraphSGAN, fake samples in the feature space of hidden layer are generated, hence it can not be applied to convolutional based classifiers. In constrast, our model generates fake nodes directly and is adaptive to convolutional based classifiers.

## 5 EXPERIMENTS

In this section, our primary goal is to show that the adversary learning can boost the performance of convolution-based GNNs in graph-based SSL under our framework. We evaluate GraphCGAN on established graph-based benchmark tasks against baseline convolution-based GNN models and some other related methods. We first introduce the dataset, experiment setup and results. Besides, we study the property of the generated nodes from our model during the training process. The ablation study is also provided in this section. The code **GraphCGAN-ICLR.zip** is provided as the supplementary file.

## 5.1 DATASETS

Three standard citation network benchmark datasets - Cora, Citeseer and Pubmed (Sen et al., 2008) are analyzed. We closely follow the setting in Kipf & Welling (2016) and Veličković et al. (2017) which allows for only 20 nodes per class to be used for training. The predictive power of the trained models is evaluated on 1000 test nodes, and 500 additional nodes are used for validation purposes.

## 5.2 EXPERIMENT SETUP AND RESULT

Two widely used of convolution-based GNNs, GCN and GAT, are considered as classifiers in GraphCGAN. In order to show the generated nodes can help improve the performance of the methods. We adopt the same model setting in the original papers (Kipf & Welling, 2016; Veličković et al., 2017). Specially, for classifier in GraphCGAN-GCN, the number of layers $L$ is 2, the dimension of the hidden layer is 16, the dropout rate is 0.5, activation function in the hidden layer is Relu. For GraphCGAN-GAT, the number of layers $L$ is 2, the dimension of the hidden layer is 8, and number

Figure 1: Ablation study for size of fake nodes $B$. It can be shown that moderate size of fake nodes can boost the classifier in graph-based SSL.

| Method | Cora | Citeseer | Pubmed |
|---|---|---|---|
| MLP | 55.1% | 46.5% | 71.4% |
| GraphSGAN (Ding et al., 2018) | $83.0 \pm 1.3\%$ | $\mathbf{73.1 \pm 1.8\%}$ | $77.2 \pm 2.6\%$ |
| DGI (velivckovic et al.,2018) | $82.3 \pm 0.6\%$ | $71.8 \pm 0.7\%$ | $76.8 \pm 0.6\%$ |
| GCN (Kipf & Welling, 2017) | 81.5% | 70.3% | 79.0% |
| GraphCGAN-GCN (ours) | $82.4 \pm 0.6\%$ | $72.6 \pm 1.0\%$ | $\mathbf{79.9 \pm 1.0\%}$ |
| Gain for GCN | 0.9 % | 2.3 % | 0.9 % |
| GAT (velivckovic et al., 2017) | $\mathbf{83.0 \pm 0.7\%}$ | $72.5 \pm 0.7\%$ | $79.0 \pm 0.3\%$ |
| GraphCGAN-GAT (ours) | $\mathbf{84.0 \pm 0.5\%}$ | $\mathbf{73.2 \pm 1.0\%}$ | $80.7 \pm 1.5\%$ |
| Gain for GAT | 1.0 % | 0.7 % | 1.7 % |

Table 1: Summary of results in terms of classification accuracy under 100 repetitions. The best and the second best results are masked in bold font. The results show that GraphCGAN-GCN and GraphCGAN-GAT outperform GCN and GAT in a significant margin, respectively.

of attention heads is 8, the dropout rate is 0.6, activation function in the hidden layer is Sigmoid. The hyper-parameter for the weight of L2 regularization is 5e-4. For the generator, we use the loss function in Equation 12 (Ablation study for loss function of generator can be found in Table 2 Appendix A). In Cora and Citeseer, we generate $B = 64$ fake nodes. In Pubmed, the number of fake nodes is $B = 256$. The ablation study of size of fake nodes are provided in Figure 1.

The results is presented in Table 1, the best and the second best results are masked in bold font. We particularly note that both GraphCGAN-GCN and GraphCGAN-GAT outperform GCN and GAT in a significant margin, respectively. More specifically, we are able to improve upon GCN by a margin of $0.9\%$, $2.3\%$ and $0.9\%$ on Cora, Citeseer and Pubmed, respectively. Besides, GraphCGAN-GAT can improve upon GAT by a margin of $1.0\%$, $0.7\%$ and $1.7\%$, suggesting that the adding fake nodes strategy in our GraphCGAN model can boost the performance for reference convolution-based GNN model. To be noticed that GraphCGAN can be easily extended to other convolution-based GNN models.

## 5.3 VISUALIZATION OF GAN PROCESS

In this subsection, we investigate about the distribution of the generated nodes during the training process. We consider three datasets to illustrate generated nodes in different perspectives. For Karate club graph (Zachary, 1977), it contains 34 nodes without features. The feature matrix $\tilde{\mathbf{X}}$ is set as identity matrix during the training process. Therefore, the plot (first row in Figure 2) of fake nodes shows the distribution of $G_2(\mathbf{z}; \tilde{\mathbf{I}})$. It can be found, after training, fake nodes mainly connect to the boundary nodes[1] which is preferred as discussed in GraphSGAN (Ding et al., 2018). MNIST datasets (LeCun et al., 1998) contain the images of handwritten digit. We can consider it as a graph

---

[1] Boundary nodes are nodes connected to different clusters

with image feature by constructing an identity adjacency matrix $\tilde{\mathbf{A}} = \tilde{\mathbf{I}}$. Therefore, the plot (second row in Figure 2) of fake feature shows the distribution of $G_1(\mathbf{z})$ which has the shape around to digit eight. Last, we generated $B = 256$ nodes for Cora dataset which are plotted in two-dimension by T-SHN (Van Der Maaten, 2014) techniques on the feature space of $\mathbf{g}(\cdot, \cdot; \theta_C)$ shown in the third row in Figure 2, which can be considered as the distribution for $G(\mathbf{z})$. We can find the generated nodes present as a complementary part for the existed nodes

Figure 2: Representation of the generated nodes during training process. **Karat club**: Real nodes are shown as colored dots representing different manually assigned groups. Three fake nodes (black) are generated, for fake nodes, the initial generated adjacency vectors are set as **1**, after training, fake nodes mainly connect to the boundary nodes; **MNIST**: One fake image as node's feature are generated, after training, the fake image shows the shape around to digit eight; **Cora**: 256 fake nodes (black) with features are generated, the plots show the t-SNE embedding for the feature space of the graph, the generated nodes show as a complementary part for the existed nodes (colored dots).

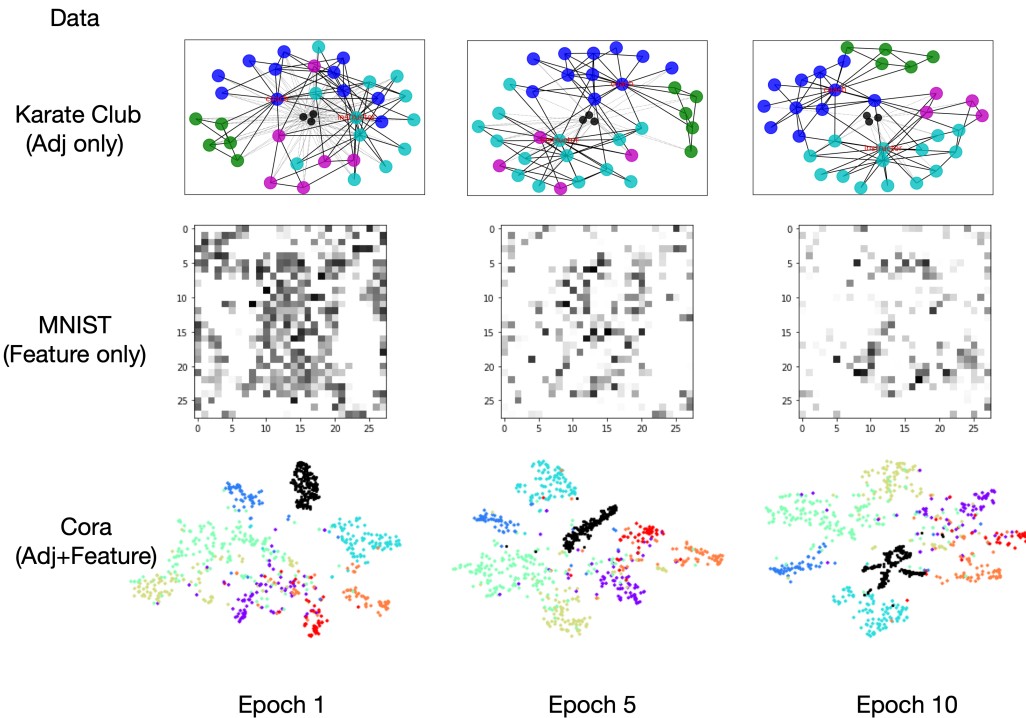

## 6 CONCLUSION

We propose GraphCGAN, a novel framework to improve the convolution-based GNN using GAN. In GraphCGAN, we design a generator to generate attributed graph, which is able to generate adjacency matrix and feature jointly. We also provide a new insight for the semi-supervised learning with convoluntional graph neural network under GAN structure. A flexible algorithm is proposed, which can be easily extended to other sophisticated architecture of GraphC, such as GAAN (Zhang et al., 2018) and GIN (Xu et al., 2018).

One potential future direction is to extend the GraphCGAN in other relevant tasks including community detection, co-embedding of attributed network (Meng et al., 2019) and even graph classification. Extending the model to incorporate edge features by generating the fake edge will allow us to tackle a larger amount of problems. Finally, in GAN, the stability of training process can be studied.

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

## A    ABLATION STUDY FOR LOSS FUNCTION OF GENERATOR

| Method | Cora | Citeseer | Pubmed |
|---|---|---|---|
| GCN (Kipf & Welling, 2017) | 81.5% | 70.3% | 79.0 % |
| GraphCGAN-GCN ($\mathcal{L}_G$) | $81.6 \pm 0.5\%$ | $71.6 \pm 0.6\%$ | $79.2 \pm 0.5\%$ |
| GraphCGAN-GCN ($\mathcal{L}_G^*$) | $82.3 \pm 0.7\%$ | $72.5 \pm 1.0\%$ | $80.0 \pm 1.1\%$ |
| GraphCGAN-GCN ($\mathcal{L}_G^{**}$) | $82.4 \pm 0.6\%$ | $72.6 \pm 1.0\%$ | $79.9 \pm 1.0\%$ |
| GAT (velivckovic et al., 2017) | $83.0 \pm 0.7\%$ | $72.5 \pm 0.7\%$ | $79.0 \pm 0.3\%$ |
| GraphCGAN-GAT ($\mathcal{L}_G$) | $82.9 \pm 0.4\%$ | $72.9 \pm 0.6\%$ | $78.9 \pm 0.5\%$ |
| GraphCGAN-GAT ($\mathcal{L}_G^*$) | $84.1 \pm 0.6\%$ | $72.9 \pm 1.1\%$ | $80.0 \pm 1.4\%$ |
| GraphCGAN-GAT ($\mathcal{L}_G^{**}$) | $84.0 \pm 0.5\%$ | $73.2 \pm 1.0\%$ | $80.7 \pm 1.5\%$ |

Table 2: Ablation study on loss function of generator.

