# OpenReview forum: "GraphCGAN: Convolutional Graph Neural Network with Generative Adversarial Networks"
_ICLR.cc/2021/Conference — Reject_

### Official Review · AnonReviewer4 · 2020-10-25
**An interesting work yet with limited significance.**

**Rating:** 5
**Confidence:** 4

**Review:**

Pros:
1.	This paper proposes the first combination of GNN with GAN for semi-supervised learning.
2.	The structure of the paper is clear and easy to follow.
Cons:
1.	Some sentences are hard to parse and many grammar errors.
2.	The contribution and novelty are limited.

This paper deals with semi-supervised learning on graphs based on GANs by proposing a framework named GraphCGAN.  The proposed framework can be easily extended to include other GNN methods. However, the novelty of the proposed model is limited, and the motivation of this paper is not strong. Also, the only motivation stated by the authors is to improve the performance on semi-supervised learning, while the improvement of the performance in the experiments is limited (i.e. ~1% improvement). The followings are the details of comments regarding this paper from three aspects.

1.	Motivation and significance are not clear.
(1)	The author(s) claim(s) that combining GCN with GANs could boost the performance of semi-supervised learning, which however is not solidly validated by the experiment results.
(2)	The author claims that GANs have never been applied to the SSL task in graphs. However, the existing method GraphSGAN [1] has already done this and GraphSGAN shows better scalability than GCN. The only difference between the proposed GraphCGAN and GraphSGAN is the selection of classifiers (MLP vs. GCN). Thus, the significance of the proposed GraphCGAN is not clear compared to the existing GCN and GraphSGAN.
As a summary, considering the limited improvement on performance, I would like to see more other motivations why we need graph-convolution-based GANs for semi-supervised learning on graphs.

       [1] Ding, M., Tang, J., & Zhang, J. (2018, October). Semi-supervised learning on graphs with generative adversarial nets. In Proceedings of the 27th ACM International Conference on Information and Knowledge Management (pp. 913-922).

2.	There are some technical issues.
(1)	The generator seems very weak. As stated in Section 3.1, the generator generates each node and edge only based on randomized latent representation “z”. If so, each new node (and edge) is generated independently without considering the dependency on the existing nodes and the dependency between the nodes and its adjacent relations.
(2)	Equation 5 is confusing. As expressed in Equation 5, the adjacent matrix of all the newly generated nodes is defined as a unit diagonal matrix I_B. It is not clear why the adjacent relation between each pair of the newly generated nodes is set to zero.
(3)	The author mentions “pull-away item” and “complementary loss” without giving any explanations on these two items in the loss function.  An ablation study is better to be conducted to validate the necessity of these two terms in loss function.

3	The presentation of the paper has many grammar errors, typos, and hard-parsing sentences.
(1)	It is hard to parse “…as the map from feature vector and adjacency vector to the space of second to the last layer in convolution-based GNN...”. It is not clear what “the space of second to the last layer in convolution-based GNN” refers to.
(2)	It is hard to parse “….and extract the map to the intermediate layer g(:; :) ….” In Algorithm 1. The term “map” is confusing.
(3)	It is not clear what "i" refers to in the equation of L_{G_2}. If it refers to the index of one node, then there may lost a “sum” symbol to sum up all the nodes into the loss.
(4)	There are many grammar errors and typos, e.g.,
-----“existed methods”
                 -----the font of word “softmax” is not unified, see Section 2.1 on Page 2.
                 -----“…Later on, Dai et al. provided a theoretical….” lacks the hyper-reference.
----- “: =”  in Equation 4.

---

> ### Author Response · Authors · 2020-11-25
> **Response to AnonReviewer4**
>
> Thank you for all valuable comments.
>
> Regarding to “Motivation and significance are not clear”:
>
> Compared to previous method, proposed method generates nodes other than hidden vectors. The generated nodes help boost the performance in state-of-the-art classifiers (GCN and its variants). The intuition can be summarized as we want to generate fake nodes to help improve the graph convolutional network and its variants.  It is important to move from intuition to a solid body of work.
>
> Regarding to “technical issue (1)":
>
> Minimizing the loss function for generator considers the dependency on the existing nodes, because the mapping from latent variable z to fake node v is shaped by the existing nodes.
>
> Regarding to “technical issue (2)":
>
> People are more interested in the classification of real nodes. Based on Equation (7), the layer propagation for real nodes is constructed based on their neighbors (both real and fake nodes). Hence, the links within fake nodes cannot affect the representation of real nodes in single propagation.
>
> Regarding to “technical issue (3)":
>
> “Pull-away item” prevents the fake nodes to shrink together. “complementary loss” leads the fake nodes to complementary space for existed nodes. We would add more explanations in next version. The ablation study for the loss function necessity can be found in Table 2 in page 10.
>
> Regarding to “grammar errors, typos, and hard-parsing sentences”:
>
> Thank you for pointing out. Most problems are addressable and could be quickly revised in our next version.

---

### Official Review · AnonReviewer1 · 2020-10-28
**incremental work combining GAN and GNN**

**Rating:** 5
**Confidence:** 5

**Review:**

This paper combines adversarial learning with graph neural networks to improve the performance of GNN for semi-supervised node classification. To generate fake nodes, the authors designed generator G1 to generate node attributes of fake nodes and G2 to generate the links of the fake nodes to existing nodes.

+ Positive
1. The idea of combining GNN with GAN for semi-supervised node classification makes sense
2. The authors provides visualization of the generated fake nodes to help understand the proposed method.
3. The proposed method is flexible, which can be used for various GNNs

- Negative
1. The novelty of the paper is limited. It is simple extension of GAN for semi-supervised learning to GNNs. The proposed method heavily relies on existing techniques with little novelty.
2. It is unclear why the generated fake nodes can only link to existing nodes, not to fake nodes. Intuitively, the fake nots are more realistic if they can also link to each other. The authors may need to give some explanations on such design.
3. Equations (9) and L_G1 are difficult to understand. For \tilde{H}_i^{L-1}, it aggregates the information of the (L-1)-hop neighborhood of x_i.  It is unclear to me why \tilde{H}_i^{L-1} can be written as g(x_i, a_i; \theta_C), i.e., only relevant to x_i and a_i. In L_G1, why do we have I_i for x_i while 0 for G_1(z). Shouldn’t both be \tilde{I}_i as we treat the adjacency matrix as I.

Question for rebuttal:
Please answer 2 and 3 above.

Justification for score：
The paper studies an important problem. However, it lacks novelty. Some part of the paper is unclear. I will change the rating if the authors can clarify the contribution and novelty, and address my questions.

---

> ### Author Response · Authors · 2020-11-25
> **Response to AnonReviewer1**
>
> Thank you for all constructive suggestions and valuable comments. The clarification of contribution and novelty could be added in our next version.
>
> Regarding to “It is unclear why the generated fake nodes can only link to existing nodes”:
>
> People are more interested in the classification of real nodes. Based on Equation (7), the layer propagation for real nodes is constructed based on their neighbors (both real and fake nodes). Hence, the links within fake nodes cannot affect the representation of real nodes in single propagation.
>
> Regarding to “Equations (9) and L_G1 are difficult to understand.”:
>
> Equation (9), \tilde{H}_i^{L-1} =  g(x_i, a_i; \theta_C), represents the L-1 Hop for node v_i, which is denoted as (x_i, a_i). We would revise as \tilde{H}_i^{L-1} =  g(v_i; \theta_C, \tilde A, \tilde X) = g(x_i, a_i; \theta_C, \tilde A, \tilde X) in next version.
>
> Similarly, For L_G1, g(x_i, I_i; \theta_c) can be revised as  g(v_i; \theta_C, \tilde I, \tilde X) =  g(x_i, I_i; \theta_C, \tilde I, \tilde X).  g(G_1(z), 0; \theta_c) can be revised as g(v_0; \theta_c, \tilde A, \tilde X) = g(G_1(z), 0; \theta_c, \tilde I, \tilde X), where we use (G_1(z), 0) to represent v_0, because (x_i, I_i) means node v_i has feature x_i and connect to i-th real node, and (G_1(z), 0) means fake node v_0 has feature G_1(z) and no connection to real node.

---

### Official Review · AnonReviewer3 · 2020-10-29
**The motivation is not strong enough that the proposed approach looks like a patchwork of two models**

**Rating:** 5
**Confidence:** 2

**Review:**

This paper proposes a novel framework to incorporate adversarial learning with convolution-based graph neural network, to operate on graph-structured data.
The proposed method is inspiring. However, some problems remain with the proposed technique.

1.	This paper announces that it proposes a novel approach called GraphCGAN to deal with the three challenges of constructing a general framework. However, the motivation is not strong enough that the proposed approach looks like a patchwork of two models.

2.	More experiments involving different scale networks are needed to prove the effectiveness of the proposed method.

This paper is globally well organized and clearly written. However, some important details are missing.

1	The details about generator are unclear.
2	The paper lacks of analysis on the experimental result.
3	The details about networks are unclear.

---

> ### Author Response · Authors · 2020-11-25
> **Response to AnonReviewer3**
>
> Thank you for valuable comments.
>
> Regarding “the motivation is not strong enough”:
>
> The work is motivated by the idea that “generated nodes can boost the performance in graph convolutional networks”. It is important to move from intuitions to a solid body of knowledge.
>
> Response to “More experiments involving different scale networks”:
>
> Pub dataset has a different scale from cora and citeseer datasets.

---

### Official Review · AnonReviewer2 · 2020-10-29
**ICLR 2021 Review of paper 2790: GraphCGAN: Convolutional Graph Neural Network with Generative Adversarial Networks**

**Rating:** 4
**Confidence:** 4

**Review:**

##### 1. Summary
The paper presents a method to combine graph convolutional neural networks (GCNs) with generative adversarial networks (GANs). The authors focus on the problem of semi-supervised learning on graphs and propose an end-to-end framework in which the generative model is followed by direct convolutions on the graph nodes. Experiments are conducted on standard benchmark datasets and the proposed method, GraphCGAN is compared against several state-of-the-art approaches.

##### 2. Rationale for the score
As stated by the authors, the proposed method is an extension of GraphSGAN. In GraphSGAN a similar process of generating fake nodes is performed. As in GraphSGAN, the fake nodes in GraphCGAN are also linked to real nodes in the graph. The difference is that in GraphSGAN a graph Laplacian regularization is done while GraphCGAN applies a convolution on the graph. This could be considered a novel change if the authors were to show that the proposed approach outperforms GraphSGAN. Yet, if there exists an increase in classification performance, it is modest at best. The accuracy and margins of error in Table 1 do not show any actual increase in performance between GraphCGAN and GraphSGAN.

##### 3. Positive aspects
- The plots included in section 5.3 are interesting and match the behavior previously reported in GraphSGAN.
- In the analysis, the authors show how their method compares to GCN and GAT when these two are included as classifiers in the end-to-end learning process of GraphCGAN. This is a good idea and the aim is to determine if there are advantages in using the generative model. Although the idea is good, there are issues with the reported values (see section 4 of this review).
- The authors included the code associated with the method. This is always appreciated.

##### 4. Negative aspects
- The experiments are not comprehensive and it is not clear if the authors performed the experiments from scratch. For example, the performance values reported for GAT and GCN are the same as the ones listed in Table 3 of [R1].
- The claim that GraphCGAN outperforms GCN and GAT is not supported by all the results in Table 1. For example, the performance on Pubmed is within the margin of error for GCN. Similarly for GAT, on all three datasets.
- When comparing against state-of-the-art methods, the authors omitted the comparison against Graph U-Nets [R1]. This is, in my opinion, the main weakness of this manuscript. If we have a look at the results obtained by Graph U-Nets (reported in Table 3 of [R1]), GraphCGAN does not seem to outperform Graph U-Nets in any dataset, including margin of errors.
- The authors refer to an ablation study but only a couple of parameters are analyzed, e.g., number of fake nodes in Figure 1 and loss function of the generator in Table 2. The many other parameters of the model are listed in section 5.2 without details about how they were derived.

##### 5. Questions to be addressed during rebuttal period
* How does GraphCGAN compare to Graph U-Nets on the Cora, Citeseer and Pubmed datasets?
* The margin of errors are quite high, similarly to those reported for GraphSGAN. Can the authors elaborate on why this is the case?
* Can the authors confirm on what set (hopefully a partition of the training set) were the plots in Figure 1 obtained? There is no margin of error associated to each point in the plots and it gives the impression of overfitting to the test set.

##### 6. Additional references
[R1] Gao, Hongyang, and Shuiwang Ji. "Graph U-Nets." ICML 2019. PMLR 97:2083-2092

---

> ### Author Response · Authors · 2020-11-25
> **Response to AnonReviewer2**
>
> Thank you for all the constructive suggestions and valuable comments.
>
> Regarding “How does GraphCGAN compare to Graph U-Nets..”:
> Thank you for pointing out literature that we were unaware of.  Graph U-Nets insert gPool and gUnpool layers to Graph Convolutional Network (GCN), which incorporates the popular technique from image processing. Our method can be extended to variants of GCN including Graph U-Nets. We conduct the experiment following the same parameter setting, the results are attached. The gains come from complementary generated nodes.
>
>
> Response to “The margin of error..”:
> The variations in our proposed method and GraphSGAN have two sources: classifier and generator. GCN and GAT methods only contain the variation from the classifier.
>
> Response to “on what set were the plots in Figure 1”:
> The Figure 1 is ablation study on size of generated nodes, the train/dev/test datasets are used following the same partition in paper [1].
>
>
> g-U-Nets                          | 84.4 ± 0.6% | 73.2 ± 0.5%    |79.6 ± 0.2%
> GraphCGAN(g-U-Nets)  |84.5 ± 0.7%  | 73.5  ± 1.2%  | 80.2 ± 1.2%
>
> Reference [1]: Kipf, Thomas N., and Max Welling. "Semi-supervised classification with graph convolutional networks." arXiv preprint arXiv:1609.02907 (2016).

---

### Decision · Program_Chairs · 2021-01-07
**Final Decision**

**Decision:**

Reject

**Comment:**

This paper presents a method to combine graph convolutional neural networks (GCNs) with generative adversarial networks (GANs) for graph-based semi-supervised learning.

**Strengths:**
  * It is a reasonable attempt to combine GCN with GAN for semi-supervised node classification.
  * The proposed method is general in that it can work with different graph neural networks.

**Weaknesses:**
  * The novelty of this work is limited.
  * The proposed method, GraphCGAN, has no significant performance improvement over state-of-the-art methods.
  * The writing has much room to improve in terms of both clarity and the linguistic quality.

Since both the novelty and the significance of this paper in its current form are limited, it is premature for publication. There is consensus among all the reviewers that this paper is not up to the acceptance standard of ICLR.